# Adaptation and pilot testing of the Early Learning Outcome Measure (ELOM) (4&5) Years Assessment tool in South African Sign Language

Alys Young[1,2]*, Claudine Storbeck[2], Katherine Rogers[1], Robyn Swannack[1,2], Andrew Dawes[3], Elizabeth Girdwood[4]

1 School of Health Sciences, University of Manchester, Manchester, United Kingdom, 2 Centre for Deaf Studies, University of the Witwatersrand, Johannesburg, South Africa, 3 Psychology Department, University of Cape Town, Cape Town, South Africa, 4 DataDrive2030, Cape Town, South Africa

* alys.young@manchester.ac.uk

## Abstract

Early Childhood Development is a key national priority in South Africa which has developed the Early Learning Outcome Measure (ELOM 4&5) specifically designed to measure the progress of 4- and 5-year-old children across 5 domains of early childhood development. This age-validated, population-standardised instrument has been shown to have measurement equivalence and lack of bias across South Africa's 11 official spoken languages. In 2023, South African Sign Language was formally recognised as 12th official language of South Africa, but no ELOM (4&5) exists in SASL despite over 6,000 deaf children being born annually. This study reports the methods used to adapt the ELOM (4&5) into SASL and establish its face validity, preliminary results from its pilot implementation with 29 deaf children and initial evaluation of its test-retest reliability and internal consistency. The small sample limits the implications that can be drawn from the results, but we report the sample performing below the Q1 marginal mean for 50–59 months olds established in the national standardisation study on all domains apart from Fine Motor Skills and Visual Motor Integration in the 60–69 months old group where the children are reaching the Q2/3 mean $(12.87 \pm 0.25)$. Seventy two percent of the sample scored in the 'falling far behind' category in comparison with 28% of children overall in South Africa. Although the ELOM (4&5)-D meets the acceptable test-retest reliability coefficient, it does not do so in all domains with wide variation between the two testing points indicating instability. Internal consistency was found not to be satisfactory. The extent to which these results are a reflection of the children's true ability, or an artefact of an underpowered sample are discussed with reference to the implications for deaf children of language deprivation in the first three years of life experienced by many deaf children in South Africa.

**Data availability statement:** The data on which this publication is based is publicly available: Young, Alys; Storbeck, Claudine; Rogers, Katherine; Swannack, Robyn; Dawes, Andrew; Girdwood, Elizabeth (2025). Early Learning Outcome Measure (ELOM) (4&5) in South African Sign Language - Dataset. University of Manchester. Dataset. https://doi.org/10.48420/28622990.v1.

**Funding:** This study was supported by a grant from the Medical Research Council (MRC) grant number: MR/V034928/1 awarded to AY and CS. Author EG is an employee of the company DataDrive 2030 who are the current owners and distributors of the ELOM 4&5 but who received no payment for this work. Author AD is an originator of the ELOM 4& 5 but received no payment for this work. The funders had no role in study design, data collection and analysis, decision to publish, or preparation of the manuscript.

**Competing interests:** Author AD is one of the originators of the ELOM 4&5 which was adapted in this research. Author EG is a senior project manager at the not for profit company DataDrive2030 who manage the licensed use of the ELOM 4&5. CS founded the not for profit early intervention programme HI HOPES from whom some of the sample are drawn.

## Introduction

Early Childhood Development (ECD) is a key national priority in South Africa (SA) [1] in line with WHO and UNICEF's definition of ECD [2,3] as both preventative of future harms, and as a practice of rights. UN Sustainable Development Goal 4.2 states the responsibility of countries to "ensure that all girls and boys have access to quality early childhood development, care and pre-primary education so that they are ready for primary education" with countries required to report on the % of children under 5 years of age who are 'developmentally on track' [4]. Additionally, in South Africa there has been the country-specific priority of addressing a significant remnant of apartheid – the major under-investment in previously disadvantaged communities. This led, in partnership with UNICEF, to the publication in 2009, by the Department of Basic Education, of the National Early Learning Development Standards (NELDS) for children birth to 4 years in South Africa [5], followed in 2015 by the National Curriculum Framework (NCF) for children birth to 4 years [6]. Although mindful of the effects of disability on early childhood development, these documents did not address any disabling condition specifically.

The NELDS and NCF were followed up by the development of the ELOM (4&5) (Early Learning Outcome Measure for 4 and 5 year olds); the first of its kind in Sub-Saharan Africa, and consistent with the constructs assessed through NELDS and NCF [7]. The ELOM (4&5) is an age-validated, population standardised instrument created in South Africa using validated measurement items in the domains of Gross Motor Development (GMD), Fine Motor Development and Visual Motor Integration (FMD & VMI), Emergent Numeracy and Mathematics (ENM), Cognition and Executive Function (ECF), and Emergent Literacy and Language (ELL). Validated for the eleven official spoken languages of South Africa with norms available for each [8], it is appropriate for children aged 50 to 69 months. It has been shown to have measurement equivalence and lack of bias across South Africa's spoken languages [8]. The results from national monitoring through the ELOM (4&5) gives a picture of the population overall and sub-groups within it who might require additional support to redress inequalities in access to early education, resources and opportunities [9]. The ELOM (4&5) is the foundational instrument in the 'Thrive by Five' triennial national index of Early Childhood Development in South Africa [10]. A recent National Planning Commission advisory document has recommended that the Thrive by Five Index be used as an indicator in the Medium Term [11].

The original validation study for the ELOM (4&5) explicitly excluded any child with a 'hearing impairment' in order not to skew the initial normative population data. The exclusion of deaf children is not unusual in the development of ECD psychometric assessments. A recent systematic review of 160 articles covering the psychometric properties of 117 ECD assessments in use in LMICs excluded all instruments that had included deaf children in their sample [12]. Furthermore, South African Sign Language (SASL) was only endorsed as an 'official language' of South Africa in 2023 [13]. Consequently, the adoption of the ELOM as part of a national policy of improvement in ECD in South Africa in effect has excluded deaf children due to the lack of an adapted ELOM (4&5) tool. The latest Thrive by Five data from 2021, and released

in 2023, on 5,222 randomly selected children includes not a single child recorded as 'having a hearing difficulty' [10]. Yet annually, over 6,000 babies are born deaf or become deaf in early childhood South Africa [14].

Unlike in high-resourced countries of the developed world, newborn hearing screening, that detects deafness in the first days of life, is not universal. The Health Professions Council of South Africa's (HPCSA) own standards of initial hearing screening - within 4 weeks and no later than 6 weeks of age, identification by 3 months, diagnosis by 4 months and early intervention begun before 6 months and no later than 8 months [15] - is later than the 1-2-3 months international best practice standards [16]. Despite this, the average age of diagnosis of childhood deafness remains at 28 months [17,18]. Only around 10% of deaf babies in South Africa are identified through newborn hearing screening [17]. The delay, by developed world standards, in identification and the start of early intervention to promote language, cognitive and socio-emotional development has deleterious consequences as comparative studies over the past 20 years in other countries have demonstrated [19].

Although some assessment and monitoring of deaf child language acquisition and development occurs in relation to specific intervention programmes in South Africa e.g. cochlear implant programmes [20] and the not for profit HI HOPES early intervention programme [17,18,21], there is little data on the majority of deaf children in South Africa, and almost nothing that addresses ECD more globally rather than just language development specifically [22]. Consequently, many deaf children arrive at school at age 5 or 6 with little information available on their individual developmental needs and, on a more structural level, no means for the state to assess the degree of need and target resources to this group to ensure optimum ECD in line with the SA's own standards and aspirations.

Working with the originators of the ELOM (4&5) [7], DataDrive2030 (the current body that trains and accredits its assessors, regulates its use and collects data from population-level surveys through the Thrive by Five Index), research studies and programme evaluations [23] and with the support of the South African Departments of Basic Education and Social Development, this study set out to adapt the ELOM (4&5) for use with deaf children who use South African Sign Language (SASL) and who make up the majority of all deaf children under 6 years old in South Africa. It also aimed to carry out an initial exploration of the internal consistency of the ELOM (4&5)-D (ELOM Deaf), establish preliminary data on its reliability through field testing, and collect pilot data on deaf children's performance.

The ELOM (4&5) required adaptation, rather than merely 'translation' into SASL because of the effects of reproducing the items in a visual (rather than spoken) language and some consequences of participant children's deafness. For example, some of the test items would not be suitable for deaf children because they are sound dependent such as the non-linguistic pencil tapping test in Domain 4 (Cognition and Executive Function) and the linguistic item in Domain 5 (Emergent Literacy and Language) which requires initial phonological discrimination. Other items if asked in a signed language would give away the answer to the child because of the visual reproduction of the answer in how a question is asked linguistically in SASL; for example, 'point to something above the table' in Item 12 (Domain 3: Emergent Numeracy and Mathematics). Other items might be potentially more confusing for a visually dominant deaf child, such as the cards in item 11 involving sorting shapes and colours if the props for this are not visually uniform (i.e. some are outlined in black and some not). Ensuring that items test the same aspect of development when implemented with a deaf child and do not disadvantage that child therefore required an evidence-based process of adaptation to establish face validity. Considerations were more akin to seeking cultural equivalence rather than linguistic equivalence, a feature usually missing in the adaption of ECD assessments [24]. For a full list of test items and more detailed description of the assessment see the ELOM (4&5) Technical Manual [23].

In what follows we present the methods of adaptation, pilot results of children's performance and initial exploration of internal consistency and test-retest reliability of the ELOM (4&5)-D.

## Methods

Additional information regarding the ethical, cultural, and scientific considerations specific to inclusivity in global research is included in the Supporting Information (S1 Checklist).

## Research aims

The overall aims of the study were:

(i) To adapt the ELOM (4&5) for use with deaf children who use SASL and to establish its face validity

(ii) To pilot-test the adapted ELOM (4&5)-D and gather preliminary data from a sample of deaf children attending a pre-school programme for deaf children

(iii) To explore the internal consistency and test-retest reliability of the ELOM (4&5)-D

## Overall design

A mixed methods study was conducted, consisting of a Delphi study for the adaption of the ELOM (4&5), pilot field testing of the ELOM (4&5)-D on a sample of deaf children and qualitative interviews with the trained assessors on their experience with the ELOM (4&5)-D. The results of these interviews will be reported separately. Data analysis explored the internal consistency and test-retest reliability of the ELOM (4&5)-D. Authors Dawes and Girwood were two of the originators of the ELOM and access to the ELOM is openly available in the ELOM Technical Manual, Appendix 1, pp. 48–53 [25]. DataDrive 2030 gave formal permission for the research study that would lead to the adaptation of the instrument into SASL and its pilot testing. The assessor training, instrument scoring and recording processes conformed to those required for the administration of the ELOM (4&5) and full details of the scoring are available in the ELOM Technical Manual [25].

## Methods

**Study 1: Delphi study.** The original ELOM (4&5) has four components all of which were assessed for potential adaptation. A: the 23 assessment items within the ELOM (4&5) Direct Assessment Manual e.g. determining whether or not the item is suitable for assessing deaf children. B: the scoring criteria including any aspects of performance that are noted e.g. for one item this includes the judgement 'did the child mumble or speak clearly?'. C: Instructions for how the assessor will present each item and tell the child what is required. D: the assessment kit which included the materials to be used in association with specific items e.g. a picture of a girl crying. Each of these components was uploaded to REDCap software (Research Electronic Data Capture) in written English for electronic access and as a common means of recording responses to the initial desk-based review and the two rounds of Delphi assessment undertaken, Fig 1.

Step One consisted of providing an initial overview of the suitability of the ELOM items for deaf children who use SASL. The study team (authors CS, RS and AY) independently reviewed components A (item) and B (scoring criteria) initially designating their suitability as yes/no/maybe. 'Yes', referred to items and their associated scoring criteria which are fine for assessing deaf children. 'No' and 'Maybe' potentially indicated either that an item and/or scoring criteria needs adaptation or potentially needs replacing with an equivalent. The responses were collated in REDCap to see points of agreement or divergence, followed by mutual discussion to agree an initial rating for all parts of components A and B. Next, components C (assessor instructions) and D (assessment materials kit) were independently reviewed by the same three members of the study team. This was to establish whether initial classifications derived from component A and B were still correct given *how* an instruction to the child would be worded for the item and the kit they are required to use. This refinement of the initial yes/no/maybe classification, in taking into consideration all four components, was classified according to 5 criteria:

(i) Items that require no modification to any of the 4 components.

(ii) Items that require no modification but their administration through SASL is potentially confounding e.g. the instruction because of the visual grammar of signed languages, gives away the answer

(iii) Items that are not suitable for deaf children because they involve an element of sound or hearing and therefore require adaptation to ensure the same aspect of ECD is assessed equivalently (e.g. rhythmic pencil tapping test)

 

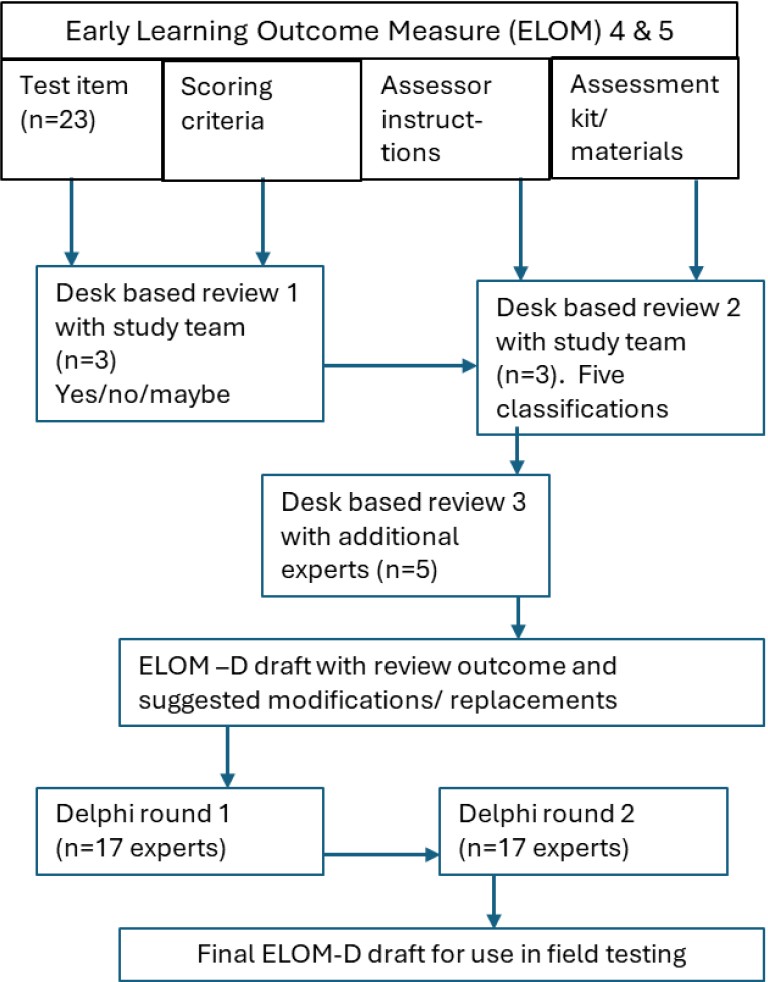

**Fig 1. Steps in the Delphi Study.**

(iv) Items that are not suitable for deaf children because they confound more than one feature e.g. auditory processing and comprehension

(v) Items that are not suitable for deaf children and cannot be adapted successfully to assess the same feature and therefore require replacement rather than adaptation.

Next a small panel of additional experts (n = 4) based in South Africa and the UK and who knew the context well were asked to consider the classifications the study team had made as a check on any internal group thinking bias that may have arisen in this process. Inclusion criteria for the additional experts were: Has 10 years or more knowledge and/or familiarity with deaf children and has one of the following professions or backgrounds: Deaf educator/classroom teacher/teaching assistant, and/or representatives from HI HOPES, a home-based early intervention program for Deaf children in South Africa (including Home Interventionist and Deaf Role Models), and/or academic researcher specialising in deaf studies, ECD or deaf children's development. This panel received all four components through REDCap with the study team ratings and a note of why the judgement had been reached. They were then asked to note whether they agreed with the classification or not and suggest any suitable modifications or replacements if indicated.

The draft ELOM (4&5)-D version in REDCap for review in the next stage (Delphi Round One) contained the conclusions of the desk-based review process on suitability and associated rationale, with some suggested replacement items or modifications. Participants were asked to 'agree', 'disagree', or state 'not sure' for all components in the adapted version and had the opportunity to add their own notes directly on to the draft ELOM (4&5)-D on REDCap for later collation. The Delphi consensus threshold was set at ≥ 70% agreement, but close attention was paid to single or anomalous responses that nonetheless might be worthy of inclusion as alternative possibilities when considering replacement items. Twenty-six experts were approached to take part in the online Delphi, 19 accepted and 17 completed the Delphi. The expert panel were drawn from varied backgrounds including Deaf educator/classroom teacher/teaching assistant, parent of deaf child(ren), specialist speech and language professional, representatives from the HI HOPES intervention team, academic researchers, member of the ELOM (4&5) originators team. All, apart from the ELOM representative, were expected to have at least 5 years' experience with deaf children. Most of the panel were based in South Africa with 3 being international. Those items with suggested modifications or replacements that were not agreed (whether in terms of test item, scoring criteria, assessor instructions or materials) were referred for discussion by the ELOM (4&5) originators with the study team and either resolved or if this were not possible, progressed to Delphi Round 2 with new suggestions of modifications and/or replacements before consensus was reached.

**ELOM (4&5)-D assessor training.** Fidelity of administration of the adapted ELOM (4&5)-D was supported by the recruitment, training and assessment to ELOM (4&5) standards of a bespoke workforce of fluent SASL users, all but one of whom were deaf. ELOM (4&5) senior trainers in collaboration with the SA members of the research team (one deaf, one hearing) devised and delivered the training through SASL with interpreters for some components. Thirteen trainees were recruited by means of purposive sampling consisting of early interventionists drawn from the HI HOPES early intervention programme in SA or working as a teacher or teaching assistant in a deaf school in SA and/or with more than 5 years direct experience of working with deaf children. The trainees had to be fluent in SASL, have reasonable competency in written English, residing in the Gauteng province of SA, with a police clearance certificate and be IT literate. The training included an introduction to the ELOM (4&5), pilot administering the ELOM (4&5)-D with at least two to three children under direct supervision and guidance of the trainer as well as the in-situ electronic scoring practice. At the end of the 5-day full time course, the training team rated the trainee's test administration according to the standard ELOM (4&5) accreditation indicators. Ten out of 13 trainees passed (9 deaf and 1 hearing) and received formal ELOM (4&5)-D assessor certification. Trainees were paid for participation in assessments at the usual rate for ELOM (4&5) assessors.

**Study 2: Pilot testing the ELOM (4&5)-D.** Following permission from Gauteng Provincial Department of Education, 6 primary schools for deaf children were approached to take part in the study which had a pre-school section. Parents of children who met the inclusion criteria were contacted by the school and supplied with a participant information sheet and consent form. The inclusion criteria, all of which had to be met, were children who are deaf or hard of hearing, who fell within the ELOM (4&5) assessment age range of 50 – 69 months, attending one of the six chosen schools and not recognised as having additional disabilities or additional needs. Following parental consent (see ethical approval below), trained and certified ELOM (4&5)-D assessors visited the school and assessed the children at a time arranged by the teacher and agreed with the school. The children were assessed twice, one week apart by the same assessor. This followed the same procedure as has been used for other test-retest reliability studies on the ELOM [26].

## Ethical approval

The study received ethical approval in South Africa from the Human Research Ethics Committee (Non-Medical), University of the Witwatersrand South Africa (H23/03/29), and further approval from Gauteng Department of Education (8/4/4/1/2). The study was also separately reviewed and approved by the University of Manchester, UK, Ethics Committee (2023-16644-29127). The recruitment period for this study ran from: 1st December 2023 to 30th April 2024. Adult assessors in this project provided written individual informed consent having read/watched a participant information sheet that was

available in SASL and written English. Written individual informed consent for participation of minors in this study was provided by their parents/guardians who received a participant information sheet in either English or SASL prior to giving consent.

**Assessor interviews.** Following the end of the assessment period, each assessor participated in a semi-structured online interview in SASL to explore their experience of using the ELOM (4&5)-D in the field, their appraisal of the effectiveness of the adaptations to the original ELOM (4&5) that have been made on an item-by-item basis, and their ideas about how the assessment process might be improved. Results of this component will be reported in the future.

## Approach to analysis

Face validity was initially established through the on-line Delphi method for each item reaching consensus. The target sample size sought to establish test-retest reliability was 49 children which is sufficient to detect an effect size of 0.50 with power set at 0.80 (p = .05) following a previous ELOM reliability study [26]. The scoring of the assessment items in the pilot-testing stage followed the ELOM (4&5) Protocol which assigns a score for each item depending on a range of criteria that exemplify the child's ability for each item e.g. numbers of attempts required or extent of completion of task. These scores are weighted depending on the item, with weightings derived from the original large scale validation study [7]. Each of the 5 domains can score a maximum of 20. Scores for each domain and total score overall are then classified according to cut-off scores [9] for 'On Track' (total and domain scores are at or above the 60th percentile of the ELOM (4&5) standard score distribution), 'Falling Behind' (total and domain scores are between the 32nd and 59th percentile of the ELOM (4&5) standard score distribution), and 'Falling Far Behind' (total and domain scores are below the 32nd percentile of the ELOM (4&5) standard score distribution). These are further differentiated according to age bracket; 50–59 months or 60–69 months old (S1 ELOM 4&5 scoring and cut offs by domain and age). Comparisons of the deaf child sample's performance were also made with the quintile marginal means established in the initial validation study of the ELOM (4&5).

Descriptive statistics (mean, median, SD, skewness and kurtosis) were calculated by domain and overall score at Times One and Two. Z-scores for each domain as well as the ELOM total score were examined to detect any outliers defined as greater than 2.58 (significant at p < .01) given this is a small sample.

The internal consistency of each of the domains of the ELOM (4&5)-D was estimated using Cronbach's α. Pearson's product moment correlation was used to explore the relationship for each of the ELOM (4&5)-D domain scores as well as the total scores at Time One and Time Two. Although the Pearson's product moment correlation usually assumes normally distributed data it has been found to be fairly robust to violations of the normal distribution and has been used in previous test-retest reliability studies of the ELOM [26]. The criterion for an acceptable ELOM (4&5)-D test-retest reliability coefficient was set at 0.75 in line with previous studies. Bootstrapping was carried out to estimate the confidence intervals, as bootstrapping does not assume any level of normal distribution. The lower and upper bound 95% confidence intervals are reported.

## Results

### Study1: The Delphi study (face validity)

Full details of the considerations raised for each item, scoring criteria, assessor instructions and assessment kit materials is the subject of a separate forthcoming article. Here we present the outcome of the Delphi process to demonstrate the initial establishment of face validity. In the following, Fig 2, 'item' refers to all four components (the test item, the scoring criteria, the assessor instructions and any associated assessment kit) and the modification and replacements that are being suggested.

### Study 2: Pilot testing

**Sample.** A convenience sample of 33 children drawn from six schools for deaf children, all in the Gauteng Province of South Africa participated in the study. Each school supplied a minimum of 2 children with one school supplying 10.

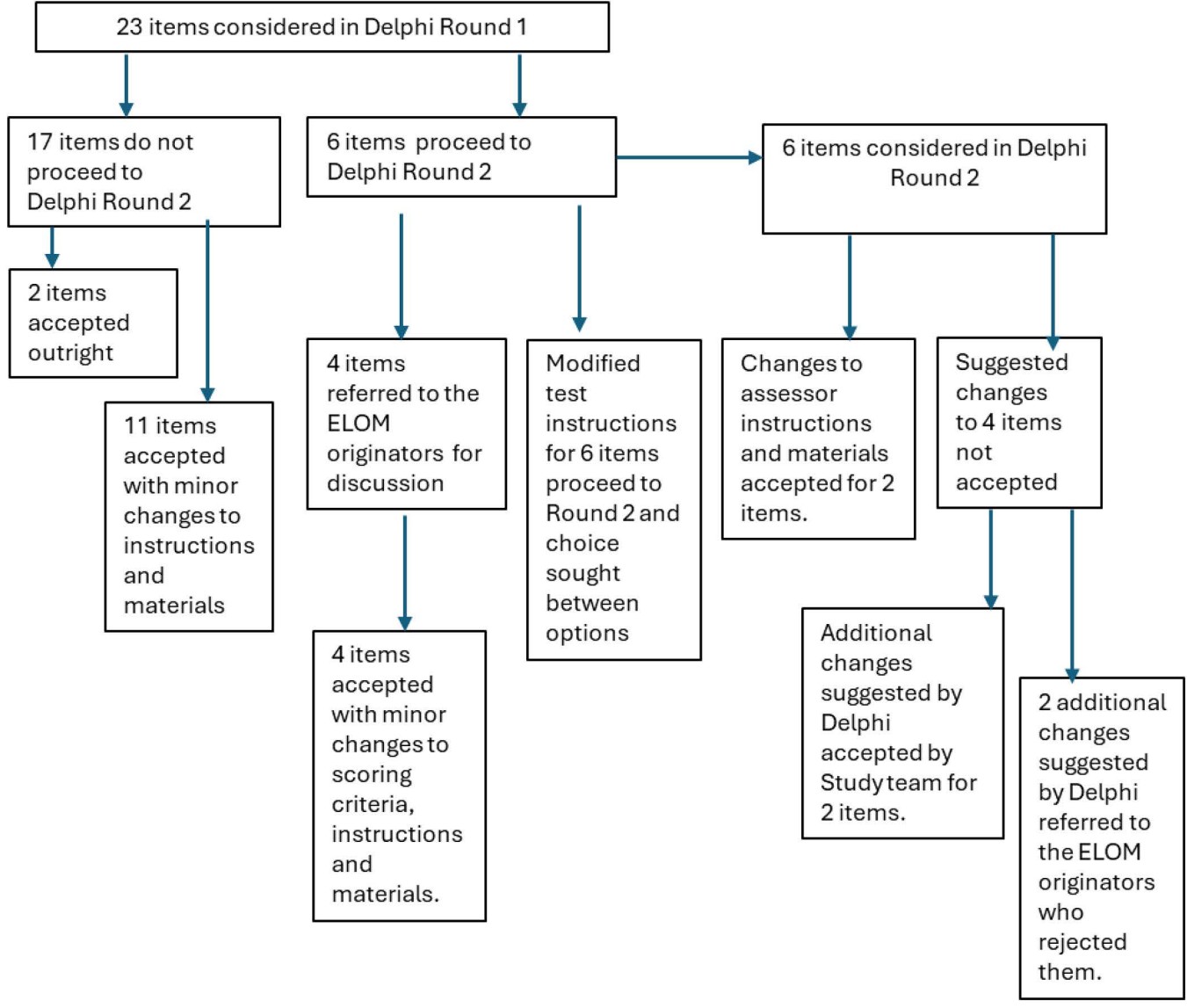

**Fig 2. Flow Chart of Delphi rounds and outcomes.**

Following data cleaning, 4 children were excluded who did not complete the ELOM (4&5)-D twice, either because they refused, or the assessor judged they were too tired to continue. The final sample size consisted of 29 children (Mean age at Time One = 61.9 months, standard deviation [SD] = 5.3, Median = 63.4; range 50.3- 69.2 months), Table 1. Nine out of the 29 children were in the age 50–59 bracket. Sixteen (16) were boys and thirteen (13) were girls. One child had two deaf parents, and another one deaf parent. Both were growing up with SASL as a home language. For the 22 children for whom data were available, the median age of confirmation of deafness was 36 months (range: 6 to 68 months). Of the 29 children, 23 had also been enrolled in the birth to six years HI HOPES early intervention programme for deaf children with more detailed data demographic available on 22 of them. Median age of enrolment in HI HOPES (n = 21) was 37 months (range 9–66 months). Median age of amplification (n = 17) was 35 months (range 11–64 months). 'Hearing age' was calculated by deducting the age or amplification from their chronological age at point of ELOM assessment (median = 18

**Table 1. Sample characteristics.**

| Sample characteristic (N = 29) | n |
|---|---|
| Sex (n = 29) | |
| Male | 16 (55.2%) |
| Female | 13 (44.8%) |
| Primary home language (n = 29) | |
| Afrikaans | 2 |
| IsiZulu | 4 |
| IsiXhosa | 4 |
| Sepedi | 5 |
| Setswana | 2 |
| Shona | 1 |
| South African Sign Language (SASL) | 2 |
| Tshivenda | 1 |
| Xitsonga | 1 |
| Unknown | 7 |
| Assessment group (n = 29) | |
| 50-59 months | 9 (31%) |
| 60-69 months | 20 (69%) |
| Degree of Deafness (n = 22) | |
| Severe | 1 |
| Severe/profound | 4 |
| Profound | 17 |
| Parents Deaf (n = 22) | |
| Yes | 2 |
| No | 20 |
| Age of confirmation of deafness (n = 22) | |
| Birth to 6 months | 2 |
| 7 months – 12 months | 3 |
| 13 months to 24 months | 3 |
| 25 months to 36 months | 4 |
| 37 months to 48 months | 7 |
| 49 months to 60 months | 1 |
| 61 months to 69 months | 2 |
| Age of amplification (n = 22) | |
| Birth to 6 months | 0 |
| 7 months – 12 months | 1 |
| 13 months to 24 months | 2 |
| 25 months to 36 months | 5 |
| 37 months to 48 months | 4 |
| 49 months to 60 months | 3 |
| 61 months to 69 months | 1 |
| No amplification | 5 |
| Hearing age (n = 17) | |
| Birth to 6 months | 3 |
| 7 months – 12 months | 3 |
| 13 months to 24 months | 3 |
| 25 months to 36 months | 4 |

*(Continued)*

**Table 1.** (Continued)

| Sample characteristic (N=29) | n |
|---|---|
| 37 months to 48 months | 3 |
| 49 months to 60 months | 1 |
| 61 months to 69 months | 0 |
| Age of entry into HI HOPES programme (n=21) | |
| Birth to 6 months | 0 |
| 7 months – 12 months | 2 |
| 13 months to 24 months | 3 |
| 25 months to 36 months | 5 |
| 37 months to 48 months | 6 |
| 49 months to 60 months | 4 |
| 61 months to 69 months | 1 |

months; range 2 – 50 months). Five out of the 22 children on which amplification data were available had never been amplified despite all being profoundly deaf.

**Descriptive statistics of assessment performance.** First, we investigated the performance of the whole sample by domain and overall, combining the scores of the nine children who are <60 months with those of the 20 who are in the older age bracket because of the small sample size; Table 2. Although the inclusion of the younger children depressed the overall means, Table 3, we did this: (i) to gain an overall picture of whether the children were performing differently between domains given that deaf children are known to have particular challenges in some areas of child development such as emergent literacy; and (ii) to start to describe the extent of homogeneity/ heterogeneity in the sample given it was a convenience sample and deaf children are well recognised to develop in typically atypical circumstances.

Skewness of distribution was acceptable except for Emergent Literacy and Language where a highly skewed positive distribution was observed (Time One: Skewness 1.86, SE.43. Time Two: Skewness 1.37, SE.43) and Gross Motor Development (Time Two: Skewness 1.25, SE.43) indicating that many of the children's scores fall very far left of the normal distribution indicating very poor performance. One child was identified as an outlier with a z score >2.58 (significant at p<.01) in the ELL domain at Times One and Two and in the GMD domain at Time Two and an overall ELOM score of 70.36 out of 100 at Time Two; a score of over double the total mean score for the sample at Time Two (34.7). The means and medians for each domain and overall increased between Time One and Time Two. It is likely that this is a practise effect whereby familiarity with the test had increased performance overall on round two. The wide interquartile range for the total scores overall (Time One: median 33.34, IQ range 18.64 – 40.30. Time Two: median 34.74, IQ range 25.58 – 42.68) shows the highly heterogenous nature of the sample.

Next, we compared the mean scores overall and by domain at Time One by age group with the marginal means by quintile established in the original standardisation study [8], Table 3 In the standardisation, after controlling for age and gender, marginal means were estimated between Quintile 1, Quintiles 2/3, and Quintiles 4/5. This comparison shows that for each age group, and in all domains, with one exception, the children are performing below the Q1 marginal mean and when age groups are combined, the overall sample is performing at the Q1 level for the 50–59 months old children. The one exception is for Fine Motor Skills and Visual Motor Integration in the 60–69 months old group where the children are reaching the Q2/3 mean (12.87±0.25).

**Performance against ELOM (4&5) cut off scores.** Using the children's total scores for Time Two only, we calculated children's performance against the cut-off scores for levels of development, Table 4. Only 2 out of the 29 (6.9%) children were 'on track' (at or above the 60th percentile ELOM (4&5) standard distribution). In one case, both parents were Deaf and SASL was a home language. In the other, both parents were hearing and IsiXhosa was the home language.

The greatest number of children have the greatest difficulty with Domain 5 Emergent Literacy and Language with 26 children (89.6%) Falling Far Behind at Time Two, followed by 17 children (59%) Falling Far Behind in Domain 3 Emergency Numeracy

**Table 2. Descriptive statistics for ELOM (4&5)-D for Times One and Two whole group combined (n = 29).**

| | GMD | | FMC & VMI | | ENM | | CEF | | ELL | | ELOM total | |
|---|---|---|---|---|---|---|---|---|---|---|---|---|
| | Time 1 | Time 2 | Time 1 | Time 2 | Time 1 | Time 2 | Time 1 | Time 2 | Time 1 | Time 2 | Time 1 | Time 2 |
| Mean | 6.69 | 7.72 | 10.91 | 12.16 | 5.60 | 6.33 | 4.91 | 5.53 | 2.18 | 3.03 | 30.29 | 34.77 |
| SD | 3.66 | 4.36 | 5.34 | 4.12 | 3.85 | 2.54 | 2.23 | 2.17 | 3.12 | 3.41 | 13.27 | 11.86 |
| Min | 1.00 | 2.91 | 2.75 | 5.86 | 0.00 | 1.00 | 0.00 | 2.26 | 0.00 | 0.00 | 6.03 | 14.93 |
| Max | 14.46 | 20.00 | 18.91 | 18.03 | 14.26 | 10.88 | 10.59 | 9.25 | 12.88 | 12.88 | 53.97 | 70.36 |
| Skewness | .09 | 1.25 | .02 | -.09 | .41 | .05 | .04 | .40 | 1.86 | 1.37 | -.16 | .74 |
| SE of skewness | .43 | .43 | .43 | .43 | .43 | .43 | .43 | .43 | .43 | .43 | .43 | .43 |
| Kurtosis | -.65 | 1.36 | -.14 | -.13 | -.47 | -.37 | .53 | -.77 | 3.71 | 1.65 | -.90 | 1.40 |
| SE of kurtosis | .85 | .85 | .85 | .85 | .85 | .85 | .85 | .85 | .85 | .85 | .85 | .85 |
| Median | 6.53 | 7.20 | 10.74 | 11.83 | 5.12 | 6.66 | 4.37 | 5.70 | 1.00 | 2.47 | 33.34 | 34.74 |
| IQ range | 3.27-9.23 | 4.58-10.02 | 5.86-16.06 | 8.53-16.66 | 2.64-8.60 | 4.85-7.85 | 3.11-6.14 | 3.81–6.64 | 0.00-3.85 | 0.00-5.17 | 18.64-40.30 | 25.58-42.68 |

**Table 3. Comparison of mean scores by age against standardisation quintiles.**

| Age and standardisation quintile | GMD | FMC & VMI | ENM | CEF | ELL | ELOM total |
|---|---|---|---|---|---|---|
| 50-59 months old (n = 9) | 4.08 | 6.50 | 3.74 | 4.52 | 1.44 | 20.28 |
| Q1 < 5 standardisation (n = 53) | 6.72 ± 1.10 | 11.88 ± 0.88 | 7.97 ± 1.10 | 4.77 ± 1.10 | 5.81 ± 1.17 | 37.15 ± 3.58 |
| 60-69 months old (n = 20) | 7.87 | 12.89 | 6.43 | 5.08 | 2.52 | 34.79 |
| Q1 > 5 standardisation (n = 61) | 8.21 ± 1.03 | 12.50 ± 0.82 | 8.08 ± 1.02 | 5.95 ± 1.02 | 6.40 ± 1.10 | 41.13 ± 3.33 |
| Both age groups combined (n = 29) | 6.69 | 10.91 | 5.60 | 4.91 | 2.18 | 30.29 |

**Table 4. Total scores by performance band and age at Time Two (n = 29).**

| | 50-59 months | 60-69 months | Total |
|---|---|---|---|
| Performance band | | | |
| On Track | 1 | 1 | 2 |
| Falling Behind | 1 | 4 | 5 |
| Falling Far Behind | 7 | 15 | 22 |

and Mathematics. At Time 2 no child is On Track for Cognition and Executive Function. Domain 2, Fine Motor Development and Visual Motor Integration, has the greatest proportion of children (n = 12, 41.0%) at Time Two who are On Track.

## Internal consistency

The internal consistency of all domains was either unacceptable or poor at Time Two with FMC & VMI and ELL deteriorating from acceptable to poor between the two testing points, Table 5.

## Test – retest reliability

Reliability of the ELOM (4&5)-D was explored using a test-retest approach based on the hypothesis that scores would be significantly correlated between the two administrations of the ELOM (4&5)-D, 7 days apart. The criterion for an acceptable ELOM (4&5)-D test-retest reliability coefficient was set at 0.75 in line with previous studies. However, the number of participants in the sample did not meet the required 49, sufficient to detect an effect 0.50 with power set at 0.80 (p = .05). Although domain FMC & VMI met the acceptability criterion, and domain ELL came close to it, the wide 95% Confidence Intervals in all domains indicate clearly that the study was under powered. It was positive to see that the ELOM (4&5)-D total result suggests reliability, however given the small sample this should be treated with caution, Table 6.

**Table 5. Internal consistency by domain at Times One and Two (n = 29).**

|  | Time One | Time Two |
|---|---|---|
| **Domain** |  |  |
| GMD (items 1–4) | α = .334 (unacceptable) | α = .581 (poor) |
| FMC & VMI (items 5–8) | α = .728 (acceptable) | α = .549 (poor) |
| ENM (items 9–13) | α = .439 (unacceptable) | α = -.455 (unacceptable) |
| CEF (items 14–17) | α = -.210 (unacceptable) | α = -.375 (unacceptable) |
| ELL (items 18–23) | α = .724 (acceptable) | α = .574 (poor) |
| Total score (items 1–23) | α = .797 (acceptable) | α = .739 (acceptable) |

**Table 6. Test-retest reliability coefficients for ELOM ((4&5))-D (n = 29).**

|  | *r* | *p* | 95% CI |
|---|---|---|---|
| GMD | .328 | >.05 | -.01,.64 |
| FMC & VMI | .841 | <.001 | .68,.96 |
| ENM | .470 | <.01 | .17,.70 |
| CEF | .600 | <.001 | .25,.82 |
| ELL | .745 | <.001 | .48, 90 |
| ELOM total | .855 | <.001 | .76,.93 |

## Discussion

This study set out to adapt the ELOM (4&5) for use with deaf children who used SASL as their primary, or only, language; to pilot the ELOM (4&5)-D in the field to gather preliminary performance data for the first time on a sample of deaf children; and to explore the test-retest reliability and internal consistency of the instrument based on these data. The Delphi consensus process involving South African and International experts established the face validity in SASL of the ELOM (4&5)-D including a reference set of instructions for each item and scoring criteria in SASL that have been approved by the originators and deaf child specific experts as matching the original intention of each item in a way that is culturally-linguistically appropriate. The smaller than expected sample recruited for this study, however, means that we are unable definitively to claim that this will be the final version of ELOM (4&5)-D without further field testing and on a much larger sample. The target sample size of 49 children for the test-retest reliability analysis sufficient to detect an effect 0.50 with power set at 0.80 (p = .05) was not achieved.

The underpowered sample is also a likely contributing factor in explaining the unacceptable or poor values in the analysis of internal consistency. There were insufficient participants given the number of items under examination for an accurate result. The intended future analysis of the semi-structured interview with the trained assessors will also provide valuable insights into whether any specific items might have proved problematic for the children to understand or were affected by any aspect of the test administration itself. Both factors could have affected the internal consistency results also.

The original plan for this research had been to recruit from 3 provinces in South Africa but one of the consequences of the Covid-19 pandemic was to restrict it to one province for travel safety reasons. Nonetheless, the difficulty in reaching target numbers was unexpected because all 6 deaf schools in the Gauteng province had agreed to participate, and all the schools had Grade R facilities. Grade R in South Africa is the reception year prior to compulsory education from 6 years old. Grade R school enrolment is optional but recommended in the Curriculum Assessment Policy Statement (CAPS) [27,28] for all children in South Africa with 95% of children in the general population attending [29]. Furthermore, all children who met the inclusion criteria took part in the study; no parent withheld their consent. The recruitment difficulties

revealed a hitherto unanticipated drop in enrolment of deaf children in Grade R in schools for the deaf for the past 5 years. Discussion with school principals suggest a range of reasons for drop in enrolment including a growth in parental perceptions of normalcy facilitated by better access to amplification technology leading to parental assumptions that specialist deaf pre-schools are not required. There has also, in Gauteng, being a proliferation of small local private pre-schools, mostly offering oral (speech) only education. These are largely inaccessible for most of the population for financial reasons but nonetheless affected numbers attending Grade R in state schools. The drop in early enrolment is of particular concern because it is well recognised that earlier access to pre-school education for deaf children is especially helpful in seeking to narrow a likely learning deficit arising from late identification of a child's deafness and language deprivation, delays and deficits that might result [15,16,17]. There are no official figures nationally on the number of deaf children enrolled in Grade R in South Africa and no deaf child was included in the latest data release of the Department of Basic Education's Thrive by Five national index of Early Childhood Development [10].

The poor performance of the sample of deaf children on the ELOM (4&5)-D is especially striking. With one exception (Fine Motor Skills and Visual Motor Integration in the 60–69 months group) the sample performed below the marginal mean for Quintile 1 established through the original standardisation by each age group, for each domain and in total. Furthermore, when the age groups were combined, the sample as a whole were performing less than the Q1 mean for the 50–59 months standardisation sample. Only 6.9% of the sample were On Track, 20.7% were Falling behind and 72.4% Falling Far Behind. This compares with the most recent Thrive by Five Index data release of 5,222 randomly selected children aged 50–59 months, using ELOM (4&5) assessment in various validated languages, in which 45.7% of children were On Track, 26.3% Falling Behind, and 28% Falling far Behind [9]. The main question raised by these results is whether the very poor performance of the sample of deaf children is a true reflection of the children's ability or whether the ELOM-D is scoring the children below their true ability? In this respect there are three pertinent issues: the small sample size, the wide variation in the children's performance within age, and the quality of their sign language in the first place to be able to understand and navigate the assessment process.

Regarding whether level of sign language development impacted on the results. The median age of confirmation of deafness in the sample was 36 months, enrolment in intervention (HI HOPES) 37 months, and age of amplification 35 months. Although these figures are consistent with other populations of deaf children in South Africa [17,18], they are far below both international standards and more significantly those set by the Hearing Professionals Council of South Africa's own statement on early intervention [15] which aims for initial hearing screening within 4 weeks and no later than 6 weeks of age, identification by 3 months, diagnosis by 4 months and early intervention begun before 6 months and no later than 8 months of age. Only 2 out of the 29 children were growing up with SASL as a home language. The rest were unlikely to have had any exposure to SASL until starting at HI HOPES or indeed exposure to any language of sufficient quantity and quality for optimal language acquisition during most of the first three years of life as they were also not able to access 'sound' until late in infancy through amplification. In other words, the effects of language deprivation [30] rather than the effects of poor SASL are more likely to be impacting their performance on the ELOM (4&5)-D. In the deaf child sample, the greatest proportion of children at Time Two Falling Far Behind (n = 26, 89.6%) was in the emerging language and literacy domain whereas in the general population sample [9] this domain had the smallest proportion of children Falling Far Behind (19.3%).

Conversely the best performing domain for the deaf child sample at Time Two was Fine Motor and Visual Integration domain with 41.4% (n = 12) On Track compared with it being the worst performing domain in the general population sample at 30.4% On Track. As the vast majority of the deaf child group of children will not have had access to sound from an early age on account of late diagnosis and late hearing aid fitting (if at all) and were growing up in families without fluent SASL, it is perhaps understandable that their visual and fine motor skills are well developed as skills not requiring language or access to sound as well as recruiting the greater visual attenuation of deaf children. This slight advantage is reinforced by the finding that Fine Motor and Visual Integration is the only performance domain reaching Q2 in comparison with the standardisation sample for the ELOM (4&5).

In terms of whether the constitution of the sample might have an effect on the poor performance of the deaf child sample, the variations between individual children's performance evidenced by the very wide interquartile ranges is of relevance. It has previously been observed that differences between deaf children are often more significant than the differences between deaf and hearing children [31] largely because they acquire language in typically atypical situations. Understanding the relationships between a range of secondary factors affecting deaf children's early childhood development including poverty and family access to early support, rather than simply the effects of deafness on language development, is a key future step in starting to target intervention for children starting school from a low level of ECD.

Exploration of the test-retest reliability scores are also helpful in unpicking whether the deaf children's performance is a reflection of their true ability or an artefact of ELOM (4&5)-D. Although overall the ELOM (4&5)-D meets the acceptable test-retest reliability coefficient, it does not do so in all domains. Although the positive correlations found were significant for 4 out of the 5 domains and overall, these results should be treated with extreme caution because of the wide confidence intervals for some domains e.g. ENM and CEF. The sample size is too small to suggest the ELOM (4&5)-D is reliable but sufficient to indicate that the SASL version is not without merit and should be tested further with a larger sample population. A larger sample population would also assist in the analysis of internal consistency which in this study was found to be unacceptable. The variation observed between the two testing points indicate an instability in the ELOM (4&5)-D. Whether this is a reflection of the underpowered nature of a highly heterogenous small sample or a true reflection that the adapted items within domains are not appropriate cannot be concluded.

## Conclusion

The principal limitation of this study is the small sample size that has reduced the power of the exploratory analyses and meant it has not been possible to meaningfully differentiate between the two age groups. Another limitation is that the SASL fluency of the sample children was not established in advance of the assessment. However, no validated test of SASL development exists and as demonstrated, the likely language deprivation of the children has perhaps more explanatory potential than their level of SASL fluency for their poor performance. Despite these uncertainties, this remains the only rigorously adapted standard instrument to explore ECD for Deaf children in sub-Saharan Africa. Although there is a strong ECD focus in LMICs with the active development of policy and practice goals that focus on 'disabled children' [32] within which deaf children are often classified, we could find no other psychometric assessment of ECD in a LMIC that had been specifically adapted and tested on a deaf population [12]. The ELOM-D (4 & 5)'s focus on deaf children within standard SA national policy ECD parameters is unique for a significant population who are persistently overlooked in national data. The recognition of SASL as the 12th official language of SA makes their inclusion imperative as SA advances its mission for greater resources and success in equalising life opportunities for its citizens through its focus on ECD. The achievement so far of a SASL instrument fit for further testing within, rather than exceptional to the national ECD index programme is significant, and further funding is being sought for a Phase Two validation constituting further field testing on a larger sample.

## Supporting information

**S1 Table. ELOM scoring and cut offs by domain and age.**
(DOCX)

**S1 Checklist. Inclusivity in global research.**
(DOCX)

## Acknowledgments

Sincere thanks to all participating schools in this study, to the Delphi panel for their time and expertise and to the study advisory board for their helpful steers.

## Author contributions

**Conceptualization:** Alys Young, Claudine Storbeck.

**Data curation:** Katherine Rogers, Robyn Swannack.

**Formal analysis:** Alys Young, Katherine Rogers, Andrew Dawes.

**Funding acquisition:** Alys Young, Claudine Storbeck.

**Investigation:** Claudine Storbeck, Robyn Swannack.

**Methodology:** Alys Young.

**Supervision:** Alys Young, Claudine Storbeck.

**Validation:** Katherine Rogers.

**Visualization:** Alys Young, Claudine Storbeck.

**Writing – original draft:** Alys Young.

**Writing – review & editing:** Alys Young, Claudine Storbeck, Katherine Rogers, Robyn Swannack, Andrew Dawes, Elizabeth Girdwood.

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
