## [Decision Letter · Decision Letter 0]

30 Jun 2025

PGPH-D-25-00807

Adaptation and pilot testing of the Early Learning Outcome Measure (ELOM) (4&5) Years Assessment tool in South African Sign Language

Dear Dr. Young,

Thank you for submitting your manuscript to PLOS Global Public Health. After careful consideration, we feel that it has merit but does not fully meet PLOS Global Public Health’s publication criteria as it currently stands. Therefore, we invite you to submit a revised version of the manuscript that addresses the points raised during the review process.

Please address the reviewers' concerns regarding methodological clarity & language use throughout the manuscript.

We look forward to receiving your revised manuscript.

Kind regards,

Avanti Dey, PhD

Staff Editor

Journal Requirements:

1. Please include a complete copy of PLOS’ questionnaire on inclusivity in global research in your revised manuscript. Our policy for research in this area aims to improve transparency in the reporting of research performed outside of researchers’ own country or community. The policy applies to researchers who have travelled to a different country to conduct research, research with Indigenous populations or their lands, and research on cultural artefacts. The questionnaire can also be requested at the journal’s discretion for any other submissions, even if these conditions are not met.  Please find more information on the policy and a link to download a blank copy of the questionnaire here: https://journals.plos.org/plosone/s/best-practices-in-research-reporting. Please upload a completed version of your questionnaire as Supporting Information when you resubmit your manuscript.

Additional Editor Comments (if provided):

Reviewers' comments:

Reviewer's Responses to Questions

**Comments to the Author**

1. Does this manuscript meet PLOS Global Public Health’s publication criteria?

Reviewer #1: Yes

Reviewer #2: Partly

2. Has the statistical analysis been performed appropriately and rigorously?

Reviewer #1: Yes

Reviewer #2: Yes

3. Have the authors made all data underlying the findings in their manuscript fully available (please refer to the Data Availability Statement at the start of the manuscript PDF file)?

Reviewer #1: Yes

Reviewer #2: Yes

4. Is the manuscript presented in an intelligible fashion and written in standard English?

Reviewer #1: Yes

Reviewer #2: Yes

Reviewer #1: The study expressed concern about the exclusion of Deaf children in the development and validation of the Early Learning Outcome Measure (ELOM) 4&5 in South Africa. In other to bridge the identified gap, the research validate the Early Learning Outcome Measure (ELOM) 4&5 among South African Deaf children who uses the South African Sign Lnaguage for communication. The research followed all necessary research ethics. Therefore publication of the manuscript when accepted is supported.

Reviewer #2: This paper describes (i) a Delphi-style adaptation of the nationally endorsed ELOM (4 & 5) school-readiness assessment for use with Deaf children whose primary language is South African Sign Language (SASL) and (ii) a small pilot study (n = 29) that explores internal consistency and test–retest reliability of the adapted instrument (“ELOM-D”). The topic is important, timely, and original: in 2023 SASL became South Africa’s 12th official language and no validated early-childhood outcome measure currently exists for Deaf children in the country. The inclusion of Deaf researchers, teachers, and language specialists in the Delphi process strengthens cultural validity. The manuscript signals a staged validation strategy, paving the way for a larger psychometric study.

The adaptation methodology is thoughtfully conceived, and the authors have taken care to meet ethical and open-science requirements (ethics approvals, prospectively registered dataset, DOI provided). Nonetheless, the psychometric evidence offered here is a bit thin to support the manuscript’s stronger claims, and some reporting gaps remain. The below suggestions will hopefully assist the authors to revise the manuscript.

In terms of the PLOS Global Public Health’s editorial criteria for publication, the manuscript attests to original research and the results of the adaptation and pilot data analysis reported have not been published elsewhere – a brief Google/Scopus check did not locate overlapping reports. The clarity of presentation and standard of English is generally good – the paper is long; trimming/condensing repetitive policy background in the Introduction would improve readability. Some specific editing changes required are mentioned at the end of this report (more are sure to be picked up with another round of careful copy/proofreading). Is there other Deaf ECD measurement work in low- and middle-income countries (e.g. Kenya, India) that could position the contribution globally?

In terms of the technical rigor and methodological detail, the paper presents an excellent outline of the Delphi process, but the final adaptation decisions are said to be presented “in a forthcoming article”; it is just noted that without those details, replication is impossible. In terms of the psychometrics, n = 29 (vs planned 49) is severely under-powered; several domains show α < 0.6 and very wide CIs on test–retest r. The authors note this throughout, but they could consider tempering conclusions further and label findings “preliminary”. The authors could consider using intraclass correlation coefficients (ICC 2,1) rather than Pearson r for test–retest reliability, as ICC incorporates systematic mean shifts, and consider adding standard error of measurement (SEM) and Bland–Altman plots to visualise agreement. Dual-site ethics approvals and parental consent are documented. The study is essentially a measurement-property study; consider completing a COSMIN checklist and provide as supplementary material. Also ensure corresponding R code / syntax for scoring is also available.

The interpretation heavily attributes low scores to language deprivation. While plausible, alternative explanations (measurement artefact, cultural unfamiliarity with some tasks, test-environment factors) could be acknowledged. A possible suggestion is to provide a short roadmap for Phase 2 validation.

The work fills a glaring gap in national ECD monitoring for Deaf children in South Africa. If strengthened, it will have clear programmatic and policy relevance and could serve as a model for other low- and middle-income country settings. My recommendation is that the manuscript deserves publication after necessary revisions.

Specific comments on the text:

- In abstract percentages oscillate between words and numerals (“Seventy two percent” vs “28 %”). Please standardise.

- Sentence in l. 55-57 needs reformulation – something is missing

- l.64 bracket is missing after ELOM

- not sure if the term “groundbreaking” is appropriate in an academic article

- sentence/clause punctuation (specifically comas, semi-colons) missing in places, e.g. l.164, 166, 167, 192, …)

- check tense consistency, e.g. l.170 included -> includes

- what does “(or similar)” mean in l.205

- quite a bit of repetition on p. 12 (l.202-230); this could be tightened up

- state Delphi consensus threshold (e.g. ≥ 70 % agreement)

- what is a “bespoke workforce”?

- l.241 be consistent with tense use

- l.252 reorder clauses

- unnecessary to report interpretation of values of Cronbach’s alpha (l.303-304)

- l.339 should this be age of amplification

- some tables require formatting attention

- l.421 has -> had

- l.442 being -> been

- l.465 replace ? with .

- l.489-494 very long sentence, something missing here

**Do you want your identity to be public for this peer review?** For information about this choice, including consent withdrawal, please see our Privacy Policy

Reviewer #1: **Yes: ** Dr Olufemi Timothy Adigun

Reviewer #2: No

---

## [Decision Letter · Decision Letter 1]

7 Aug 2025

Adaptation and pilot testing of the Early Learning Outcome Measure (ELOM) (4&5) Years Assessment tool in South African Sign Language

PGPH-D-25-00807R1

Dear Professor Young,

We are pleased to inform you that your manuscript 'Adaptation and pilot testing of the Early Learning Outcome Measure (ELOM) (4&5) Years Assessment tool in South African Sign Language' has been provisionally accepted for publication in PLOS Global Public Health.

Best regards,

Julia Robinson

Executive Editor

Reviewer Comments (if any, and for reference):

Reviewer's Responses to Questions

**Comments to the Author**

Reviewer #1: All comments have been addressed

Reviewer #2: All comments have been addressed

publication criteria?

Reviewer #1: Yes

Reviewer #2: Yes

3. Has the statistical analysis been performed appropriately and rigorously?

Reviewer #1: Yes

Reviewer #2: Yes

4. Have the authors made all data underlying the findings in their manuscript fully available (please refer to the Data Availability Statement at the start of the manuscript PDF file)?

Reviewer #1: Yes

Reviewer #2: Yes

5. Is the manuscript presented in an intelligible fashion and written in standard English?

Reviewer #1: Yes

Reviewer #2: Yes

Reviewer #1: I have read through the manuscript all over again. The authors have significantly improve on the earlier submission. Unfortunately, the author did not add the table of correction with the corrected submission thereby making it difficult to actually pinpoint the corrections made at a glance. Otherwise, the manuscript is worthy of progress as it present sailit issues in Deaf education in South Africa.

Reviewer #2: I am satisfied that the authors have addressed all the comments or indicated why changes have not been made, with appropriate argumentation.

**Do you want your identity to be public for this peer review?** For information about this choice, including consent withdrawal, please see our Privacy Policy

Reviewer #1: **Yes: ** Olufemi Timothy Adigun

Reviewer #2: No
